# MT$^2$: Multi-task Mean Teacher for Semi-Supervised Cell Segmentation

**Binyu Zhang**[1]
School of Artificial Intelligence, BUPT
Beijing, China 100876
`zhangbinyu@bupt.edu.cn`

**Junhao Dong**[1]
School of Artificial Intelligence, BUPT
Beijing, China 100876
`djh1999@bupt.edu.cn`

**Zhicheng Zhao**[*]
School of Artificial Intelligence, BUPT
Beijing, China 100876
`zhaozc@bupt.edu.cn`

**Zhu Meng**
School of Artificial Intelligence, BUPT
Beijing, China 100876
`bamboo@bupt.edu.cn`

**Fei Su**
School of Artificial Intelligence, BUPT
Beijing, China 100876
`sufei@bupt.edu.cn`

## Abstract

Cell segmentation is significant for downstream single-cell analysis in biological and biomedical research. Recently, image segmentation methods based on supervised learning have achieved promising results. However, most of them rely on intensive manual annotations, which are extremely time-consuming and expensive for cell segmentation. In addition, existing methods are often trained for a specific modality with poor generalization ability. In this paper, a novel semi-supervised cell segmentation method is proposed to segment microscopy images from multiple modalities. Specifically, Mean Teacher model is introduced to a multi-task learning framework, named Multi-task Mean Teacher (MT$^2$), in which both the classification and the regression heads are utilized to improve the prediction performance. Moreover, new data augmentation and multi-scale inference strategies are presented to enhance the robustness and generalization ability. For the quantitative evaluation on the Tuning Set of NeurIPS 2022 Cell Segmentation Competition, our method achieves the F1 Score of 0.8690, which demonstrates the effectiveness of the proposed semi-supervised learning method. Code is available at `https://github.com/djh-dzxw/MT2`.

## 1   Introduction

Microscopy images are widely used in biology and biomedical research, which are significant to the analysis and understanding at the cellular level [1]. With the advances in microscopy, large-scale high-quality cell images can be easily obtained. However, the management and analysis of these images are always time-consuming and costly. In this case, cell segmentation techniques [2], such as thresholding [3] and region-growing [4] were proposed to locate the regions of interest automatically, thereby laying a solid foundation for subsequent analysis and processing, such as cell tracking [5].

---

[*]Corresponding Author
[1]These authors contributed equally.

36th Conference on Neural Information Processing Systems (NeurIPS 2022).

In recent years, image segmentation methods based on deep learning in medical field have developed rapidly. The cell segmentation task aims to detect and delineate each distinct object of an image, and the mainstream methods can be divided into two categories. The first branch was based on object detection, such as Mask R-CNN [6] and DetectoRS [7], where the bounding box of each cell was generated firstly, and then its boundary was demarcated. However, due to the shape diversity and dense distribution of cells in the microscopy images, the bounding boxes were often too coarse to provide precise localization, and ultimately led to false predictions. The other branch was based on semantic segmentation, such as SegNet [8] and DeepLab [9], which achieved high accuracy on benchmark datasets. Accordingly, a slice of medical image segmentation methods [10][11][12][13] were also improved. However, most of them depended on intensive and costly annotations labeled manually. Therefore, semi-supervised semantic segmentation models [14][15][16][17] were proposed to utilize a large amount of unlabeled data with merely limited labeled one. However, most existing methods were trained on a specific distribution of data, thereby resulted in a poor robustness and generalization capability in dealing with various distributed data. Furthermore, instead of predicting the class of each pixel, cell segmentation needs to further distinguish which pixels belong to the instances. Hence, an additional procedure is required to convert the semantic segmentation maps into the instance ones.

In this work, we propose a novel semi-supervised cell segmentation framework, named Multi-task Mean Teacher ($MT^2$), which is fit to process multi-modality high-resolution microscopy images. It promotes the Mean Teacher [14] by applying multi-task learning. Particularly, two auxiliary heads, i.e., regression head and classification one, are jointly trained to determine cell boundaries. Besides, to handle the images of multiple modalities, a variety of strong data augmentations and a multi-scale inference strategy are used for better generalization ability. Moreover, three public external datasets along with an effective post-processing algorithm are also leveraged to obtain more accurate instance segmentation results. Our main contributions are summarized as follows:

- · A novel semi-supervised cell segmentation framework is proposed based on the Mean Teacher model and a multi-task learning strategy.

- · Strong data augmentations (especially domain-agnostic augmentations) and a multi-scale inference are utilized to ensure the robustness and generalization across multiple modalities.

- · An effective post-processing algorithm is developed to achieve the promising performance. It is further strengthened by the application of three extra datasets.

- · Evaluation on Tuning Set of NeurIPS 2022 Cell Segmentation Competition [18] has demonstrated the remarkable effect and generalization of our model.

The remaining sections are organized as follows: In Section 2, the proposed method is elaborated, including data pre-processing, network architecture, semi-supervised strategy and post-processing. Details of the experiments are introduced in Section 3, and we present the discussion and analysis in Section 4. Lastly, a conclusion of our work is confabulated in Section 5.

## 2 Method

In this section, we propose a novel cell segmentation method based on semi-supervised learning, named Multi-task Mean Teacher ($MT^2$) for multi-modality microscopy images. The whole pipeline can be separated into 3 stages, including pre-processing, deep learning model and post-processing.

### 2.1 Pre-processing

Since the dataset contains images of multiple modalities with different distributions, a simple but effective pre-prossessing method is designed to normalize images. Firstly, all the images are transferred into 3 channels. Specifically, we repeat those images only having 1 channel, such as phase-contrast and differential interference contrast images, 3 times and stack them together. We also expand 2-channel fluorescent images into 3 channels by filling the third channel with zeros. Secondly, the pixel values in the images are normalized into [0, 255].

## 2.2 Multi-task Mean Teacher (MT²)

The overview of our method is illustrated in Figure 1. The proposed method MT² is based on Mean Teacher Model [14] with a prevalent segmentation network. The teacher and student model share the same network architecture with two auxiliary regression heads to achieve a multi-task learning objective. The labeled data are only passed to the student model with strong augmentation, while the unlabeled data are passed to both teacher and student model with strong and weak augmentation respectively to conduct a semi-supervised learning.

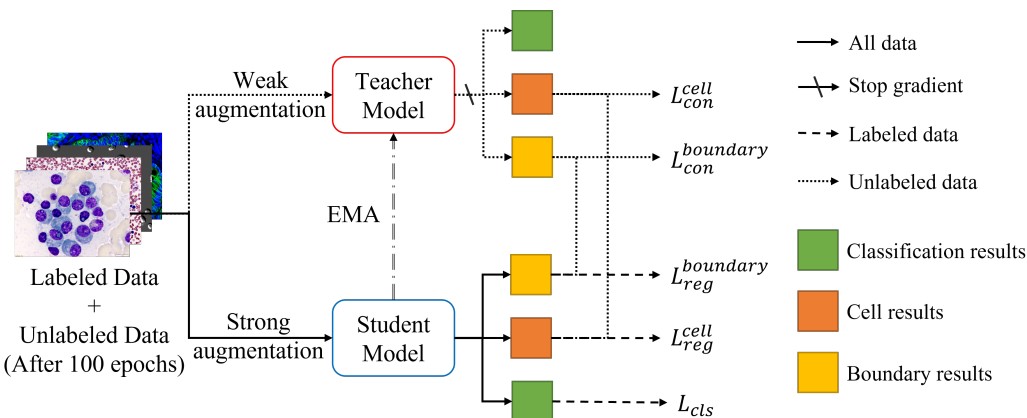

Figure 1: The overview of the proposed MT² framework.

### 2.2.1 Network architecture

Considering the dense distribution of cells in the images, as well as the diversity of image modalities and scales, densely feature fusion and densely skip-connections are needed on different network levels to obtain more capable features. Consequently, we utilize segmentation network UNet++ [19] as the backbone for the student model and teacher model in our framework, shown in Figure 2. UNet++ simultaneously leverages long and short connections, which can capture features at different levels and integrate them through directly stacking. Therefore, the combined features always have smaller differences leading to a more accurate prediction. After the feature extractor, 3 prediction heads, i.e., a classification head, a cell head and a boundary head are linked and each prediction head has a unique task. For the classification head, it is utilized to classify every pixels into 3 predefined classes, which are background, cell and boundary. Note that the boundary class is added to separate the adjacent cells, since we treated this instance segmentation task as a semantic segmentation way, and consequently it's very possible that several adjacent cells are predicted as one cell.

Unfortunately, due to the number of pixels is extremely unbalanced between cell and boundary class, the classification head is still difficult to predict the accurate boundary. As a consequence, apart from the classification head, a cell head and a boundary regression head are also applied to predict the cell and boundary heatmaps individually. The pixel values of both predicted heatmaps range from 0 to 1, which represents the probability that each pixel belongs to a cell or boundary. This multi-task strategy is confirmed to be effective because more supervisions are leveraged in the training procedure.

As for the supervision information for the cell head and boundary one, we generate the cell and boundary target heatmaps by the following steps. Firstly, the boundary of each cell can be generated using the instance labels. Secondly, the cell radius $r_i$ can be obtained by generating a distance map, which measures the distance of the pixels in the cell to it's boundary. After that, for each cell instance, the pixels in cell target heatmap whose corresponding pixels values in the distance maps are larger than $\frac{r_i}{5}$ are set to 1, while others are set to 0. And in the boundary target heatmaps, only the boundary pixels are to 1 and others are kept as 0. Finally, a Gaussian blur are conducted to each cell's heatmaps, of which the kernel size for cell heatmap is set as $\frac{r_i}{5}$ and $\frac{r_i}{10}$ for boundary heatmap. So that, two heatmaps can be obtained for each image, whose pixel values are in [0, 1], one represents the cell area and the other represents the boundary area.

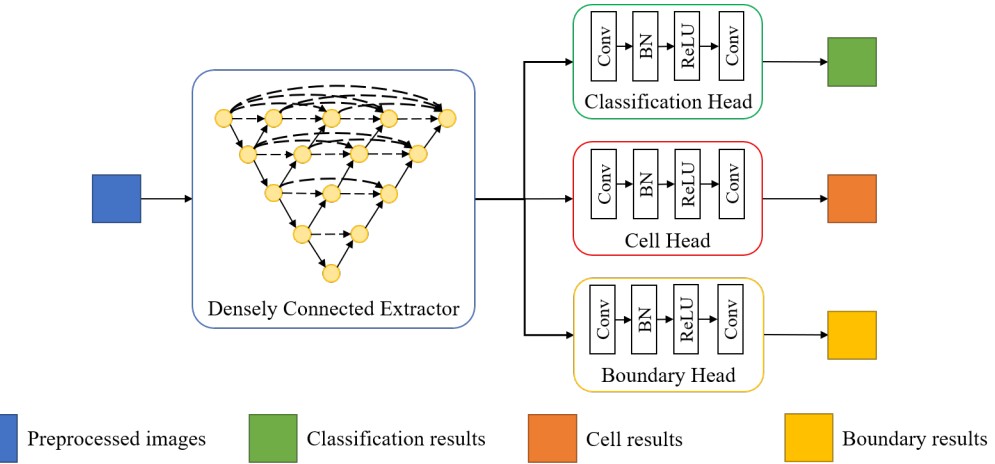

Figure 2: The architecture of our network with UNet++ as a base feature extractor.

### 2.2.2 Semi-supervised learning

The overall process is also shown as Figure 1. For labeled data, to supervise the student model from different level, we use a compound loss function $L_{cls}$ for classification head,

$$L_{cls} = L_{ce}(\hat{Y}_{cls}, Y_{cls}) + L_{dice}(\hat{Y}_{cls}, Y_{cls}) \tag{1}$$

where $L_{ce}$ denotes the cross entropy loss and $L_{dice}$ represents the Dice loss, $\hat{Y}_{cls}$ and $Y_{cls}$ stand for the class predictions and labels of the images respectively. The cross entropy loss can supervise the predictions in pixel level. The Dice loss can evaluate the Intersection over Union (IoU) of predictions in global level and has good performance for scenes with serious imbalance of positive and negative samples. For cell head and boundary head, a compound loss function is also applied as $L_{reg}$,

$$L_{reg}^X = L_{ssim}(\hat{Y}^X, Y^X) + \lambda L_{mse}(\hat{Y}^X, Y^X) \tag{2}$$

in which $L_{mse}$ indicates the MSELoss function and $L_{ssim}$ is the SSIMLoss [20] function, $\hat{Y}^X$ and $Y^X$ ($X$ chosen from cell or boundary) denote the cell or boundary predictions and labels of the images. MSELoss enables pixel-wise supervision of predictions, while SSIMLoss can perform local structural supervision of predictions. The hyper-parameter $\lambda$ is used to balance the weight between the two loss functions, and we set $\lambda = 10$ in the experiments. So the total loss function for labeled data can be summarized as:

$$L_{labeled} = L_{cls} + L_{reg}^{cell} + L_{reg}^{boundary} \tag{3}$$

In order to use unlabeled cases, we introduce Mean Teacher [14] model. Specifically, the student model ($S$) and the teacher model ($T$) have the same network architecture. Only the gradient of $S$ is calculated to update parameters, while $T$ is updated through Exponential Moving Average (EMA) [14], i.e., the temporal ensemble of $S$. Hence, the prediction of $T$ should be less noisy. Each batch of unlabeled images is duplicated and applied strong and weak augmentations individually. The teacher model $T$ predicts the weakly augmented images, while $S$ predicts the strongly augmented ones. The adopted consistency loss $L_{con}$ is as follows,

$$L_{con}^X = L_{mse}(\hat{Y}_S^X, sg(\hat{Y}_T^X)) \tag{4}$$

where $\hat{Y}_S^X$ and $\hat{Y}_T^X$ represent the predictions of the student and teacher model respectively, and $sg(\cdot)$ is the stop gradient function to prevent the gradients from being passed back to the teacher model. This consistency loss is applied on both cell and boundary heatmaps to ensure the prediction of $S$ is closer to that of $T$.

## 2.3 Post-processing

Although the predicted pixel-wise classification, cell and boundary results can be obtained via MT$^2$ framework, they only represent the probability that each pixel belongs to the foreground or background. These results also cannot illustrate which cell each pixel belongs to. Moreover, it is likely that many cells gather in a local region, which will easily lead to wrong segmentation if there is only one threshold used to separate foreground and background.

Among three types of predictions, the cell and the boundary results are chosen for their better performance. To separate the gathered cells, the boundary results are subtract from the cell results. Considering the characteristics of the cells, in light of CISCNet [21], the watershed algorithm [22] is adopted in the post-processing procedures, which are shown in Figure 3. First, two thresholds $T_{cell}$ and $T_{seed}$ are set for the algorithm (shown as Figure 3a). The pixels whose values higher than $T_{cell}$ are considered as foreground mask, which limits the scope of subsequent dilation, while $T_{seed}$ separates the gathered cells and leaves each one a seed for watershed (e.g., the red circle in Figure 3a). Then, each seed whose area is smaller than $S_{min}$ is discarded to filter out the prediction noise such as false detection (e.g., yellow circles in Figure 3b). After that, the remaining seeds are labeled with unique numbers, which represent the cell instance IDs (shown as Figure 3c). At last, dilation algorithm is implemented on these seeds with the target of the foreground masks, and watershed is leveraged to determine the final boundaries of the cell segmentation. Figure 3d demonstrates the final segmentation result, in which foreground masks with no seeds are removed. In the implementation, we set $T_{cell} = 0.3$, $T_{seed} = 0.8$ and $S_{min} = 64$, which has been proven to be effective on the tuning set.

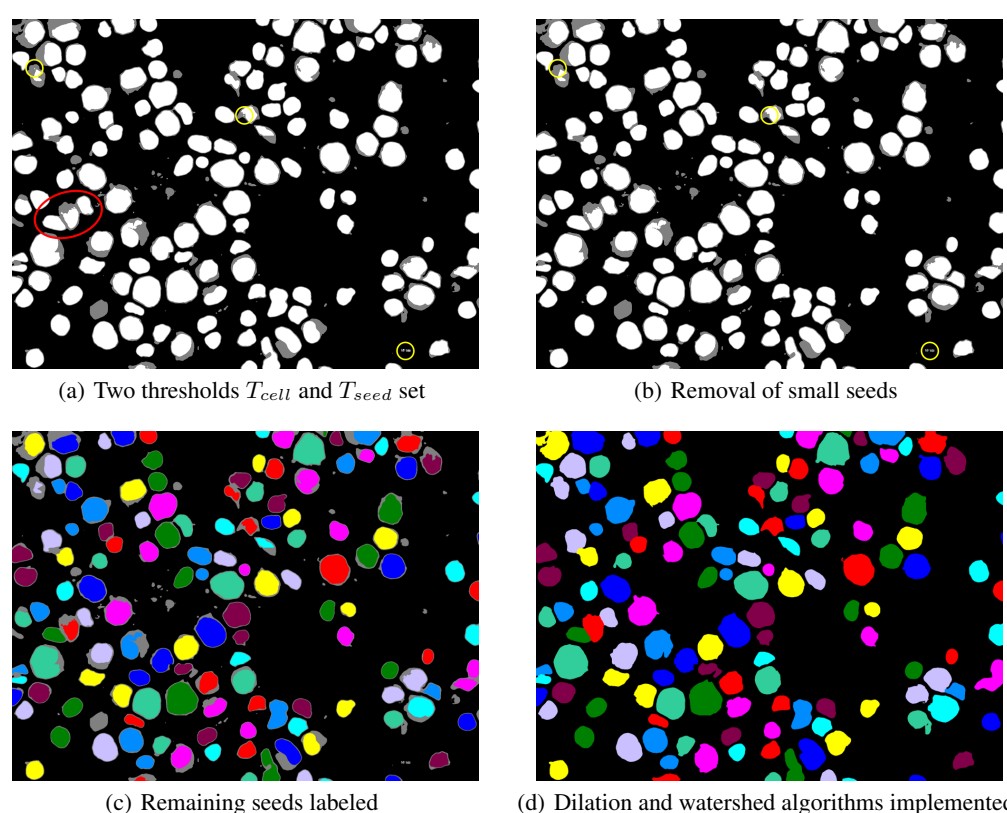

(a) Two thresholds $T_{cell}$ and $T_{seed}$ set          (b) Removal of small seeds

(c) Remaining seeds labeled          (d) Dilation and watershed algorithms implemented

Figure 3: Procedures of the post-processing algorithm.

# 3 Experiments

## 3.1 Dataset

The official dataset contains 1000 labeled and 1713 unlabeled images from various microscopy types, tissue types, and staining types for training, 101 images for validation and 200+ images for test. There are four microscopy modalities in the training set, including brightfield (300 patches), fluorescent (300 patches), phase-contrast (200 patches) and differential interference contrast (200 patches). Moreover, we add three public external datasets, including Omnipose [23], Livecell [24] and Tiussenet [25]. Omnipose dataset contains 4833 images constituting approximately 700,900 individual cells deriving from 14 bacterial species. Livecell consists of more than 1.6 million annotated cells of eight morphologically distinct cell types, grown from early seeding to full confluence, and has undergone rigorous quality assurance to minimize bias in the annotations. Tissuenet is a large comprehensive segmentation dataset of >1 million paired whole-cell and nuclear annotations.

## 3.2 Implementation details

### 3.2.1 Environment settings

The development environments and requirements are presented in Table 1.

Table 1: Development environments and requirements.

| System | Ubuntu 18.04.4 LTS |
|---|---|
| CPU | Intel(R) Xeon(R) CPU E5-2698 v4 @ 2.20GHz |
| RAM | 503GB; 2667MT / s |
| GPU (number and type) | Four NVIDIA Tesla V100 32G |
| CUDA version | 10.2 |
| Programming language | Python 3.7 |
| Deep learning framework | Pytorch (Torch 1.8.0, torchvision 0.9.0) |
| Specific dependencies | None |
| Code | `https://github.com/djh-dzxw/MT2.git` |

### 3.2.2 Training and inference protocols

**Traing in details.** In order to improve the inference speed while maintaining the accuracy, ResNet-50 [26] pretrained with ImageNet [27] is used as the backbone of feature extractor and the weights of two extra heads are initialed randomly. We adopt AdamW as the optimizer and employ a cosine annealing learning rate scheduler to train our model.

For the supervised learning on the labeled data, a variety of data augmentation methods are used to improve the robustness. We only use fully-supervised learning in the first 100 epochs as warm-up stage, and apply the teacher model in the next 400 epochs. The initial learning rate is set to 0.001 and the weight parameter $\beta$ of EMA is set to 0.995 in all of our experiments. More details are demonstrated in Table 2.

**Data augmentation.** For labeled cases, a variety of data augmentation methods are adopted. For each training sample, the image is rescaled with a random scale factor from 0.5 to 2 and randomly cropped to a $512 \times 512$ patch. Then, randomly flip and rotation are applied the patch. Due to great differences in the images of various modalities, colorjitter and cutmix are utilized to augment the training data. Note that, for the brightfield images, the stained cells are the only type of the cell need to be segmented, instead of all the cells. And color is one of the most important clues to distinguish them, so the random saturation and random hue operations in the colorjitter are harmful to prediction. Therefore, only the random brightness and random contrast are adopted in the colorjitter process.

For unlabeled images, we duplicate each batch of them and apply weak and strong augmentation respectively. The weak transformation consists of random scale, cropping, flipping and rotation, while the strong one adds colorjitter and cutmix.

**Inference in details.** During inference phase, since the cell boundaries of the fluorescence images are very blurred and differ greatly from other images, we process these images individually. For the

Table 2: Training protocols of the proposed method.

| Model | Student | Teacher |
|---|---|---|
| Network initialization | ImageNet pretrained weight | Student trained with 100 epochs |
| Batch size | 24 | 24 |
| Patch size | 512×512 | 512×512 |
| Total epochs | 500 | 400 |
| Optimizer | AdamW | EMA ($\beta = 0.995$) |
| Initial learning rate (lr) | 0.001 | None |
| Lr decay schedule | CosineAnnealingLR | None |
| Training time | 46 hours | 40.5 hours |
| Loss function | $L_{cls}, L_{reg}$ | $L_{con}$ |

fluorescence images, which can be identified by their number of channels, we scale them up by 1.5 times, while for others, a multi-scale inference is utilized with three scale factors (i.e., $0.8\times$, $1.0\times$ and $1.2\times$) to rescale the images. Next, these scaled images are fed into the model in a sliding-window inference fashion to obtain the predictions. After that, the predictions of scaled images are rescaled and fused at their original size. Specifically, for each pixel, we take the maximum value of the corresponding pixels in all the predictions as the final results.

## 4 Results and discussion

### 4.1 Evaluation metrics

All the experimental results are evaluated by two indicators: F1 score and running time. F1 score is a metric to evaluate the performance of a model, considering the precision and recall at the same time.

$$F_1 = \frac{2 \cdot P \cdot R}{P + R}, \tag{5}$$

where $P$ and $R$ represent the precision and recall of the model predictions, respectively. And for the precision and recall of the model, we utilize IoU for calculation. If a predicted instance from the model and a target instance from the ground truth have an IoU larger than 0.5, it is said that the prediction hit the target, and vice versa. The running time indicator is a metric to evaluate the speed of a model. In the experiments, all the testing images are evaluated one by one. To compensate for the Docker container startup time, there is a time tolerance $T_{tolerance}$ for the running time.

$$T_{tolerance} = \begin{cases} 10s, & if\, H \times W \leq 10^6; \\ \frac{H \times W}{10^6} 10s, & if\, H \times W > 10^6. \end{cases} \tag{6}$$

in which $H$ and $W$ represent the height and width of the images. And the running time is calculated as $T_{running}$ with the actual time $T$ every prediction taken,

$$T_{running} = max\{0, T - T_{tolerance}\}. \tag{7}$$

### 4.2 Quantitative and qualitative results on tuning set

For quantitative evaluation on the tuning set, our proposed method has achieved the F1 Score of 0.8690 on the leaderboard, which shows the high accuracy and generalization across multiple modalities of medical images. For supervision mode, we obtain the score of 0.8148, which is 0.0175 lower than the semi-supervised setting with the same training data. It demonstrates that the unlabeled cases actually improve the performance of the segmentation network.

To explore the effectiveness of our method, we conduct ablation experiments on the tuning set. In Table 3, Reg indicates if the cell and boundary heads are attached to the network. Post, CM, MS, CJ and SS represent if post-process, cutmix, multi-scale inference, colorjitter and semi-supervised methods are applied respectively. "✓✓" in the MS column represents we especially use 1.5 as scale

factor to the fluorescence images. And Ext data shows the extended datasets, in which "O" for Omnipose [23], "L" for Livecell [24] and "T" for TissueNet [25].

Table 3: The ablation study of our model.

| Model Arch. | | | Data Aug. | | | Inference | | F1 score |
|---|---|---|---|---|---|---|---|---|
| Base Network | Reg | SS | CM | CJ | Ext data | Post | MS | |
| UNet | | | | | | | | 0.5375 |
| UNet | ✓ | | | | | | | 0.5557 |
| UNet | ✓ | | | | | ✓ | | 0.7309 |
| UNet | ✓ | | ✓ | | O | ✓ | | 0.7350 |
| UNet | ✓ | | ✓ | | O | ✓ | ✓ | 0.7435 |
| UNet | ✓ | | ✓ | | O+L | ✓ | ✓ | 0.7522 |
| UNet++ | ✓ | | ✓ | | O+L | ✓ | ✓ | 0.7865 |
| UNet++ | ✓ | | ✓ | ✓ | O+L | ✓ | ✓ | 0.8148 |
| UNet++ | ✓ | ✓ | ✓ | ✓ | O+L | ✓ | ✓ | 0.8323 |
| UNet++ | ✓ | ✓ | ✓ | ✓ | O+L+T | ✓ | ✓ | **0.8558** |
| UNet++ | ✓ | ✓ | ✓ | ✓ | O+L+T | ✓ | ✓✓ | **0.8690** |

In order to explain how semi-supervised learning improves the performance, Figure 4 is introduced to illustrate the comparison of segmentation results between the supervised-only (SO) and semi-supervised (SS) models. It can be seen that the SS method is more accurate and robust. First, more positive samples can be predicted. Cells obtained in Figure 4c is obviously more than those in Figure 4b, especially for the ones with less significant features. For example, in the first row, samples pointed by the red arrows are ignored by the SO model, but the SS model can still make accurate predictions. Second, the predictions usually have finer cell boundaries, especially in the second and last rows. Third, fewer false positives (FPs) are generated. For instance, the yellow arrows in the first row indicate the regions where FPs occur in Figure 4b, but not in the predictions of the SS model.

There are several visualized examples of successful (c and d) and failed cases (a and b) shown in Figure 5. In the experiments, we find the failed predictions can be mainly attributed to three reasons. First of all, there are variety shapes of cells, but samples from each kind of cells are unbalanced, especially the rod-shaped bacteria images. So that the model will suffer from under- or over-segmentation problems (e.g., Figure 5a). To address this issue, we apply unlabeled data in a semi-supervised manner and add extend datasets. Second, some images are of poor quality and each cell boundary is very blurred, such as fluorescence images, so it is difficult to effectively find the dividing lines between cells (e.g., Figure 5b). Hence, in the inference phase, we purposely apply a larger scale factor to the fluorescence images to alleviate the problem. Third, for the same kind of cells, their color and size are also quite different. So we utilize the strong and weak data augmentation to the data, including random scale, cutmix and colorjitter. Furthermore, we can observe that images of cells with sharp edges tend to have better segmentation results (e.g., Figure 5c and 5d). Finally, we rank the 4th place on the tuning set in NeurIPS 2022 Cell Segmentation Competition as shown in the Table 4.

Table 4: The rank of $MT^2$ on the tuning set.

| Team | F1 score |
|---|---|
| osilab | 0.9067 |
| cells | 0.9004 |
| Sribd-Med | 0.8809 |
| BUPT-MCPRL (Ours) | 0.8690 |
| vipa | 0.8537 |
| RedCat_AutoX | 0.8535 |
| cphitsz | 0.8428 |
| saltfish | 0.8250 |
| Overoverfitting | 0.8122 |
| train4ever | 0.8097 |
| naf | 0.7740 |

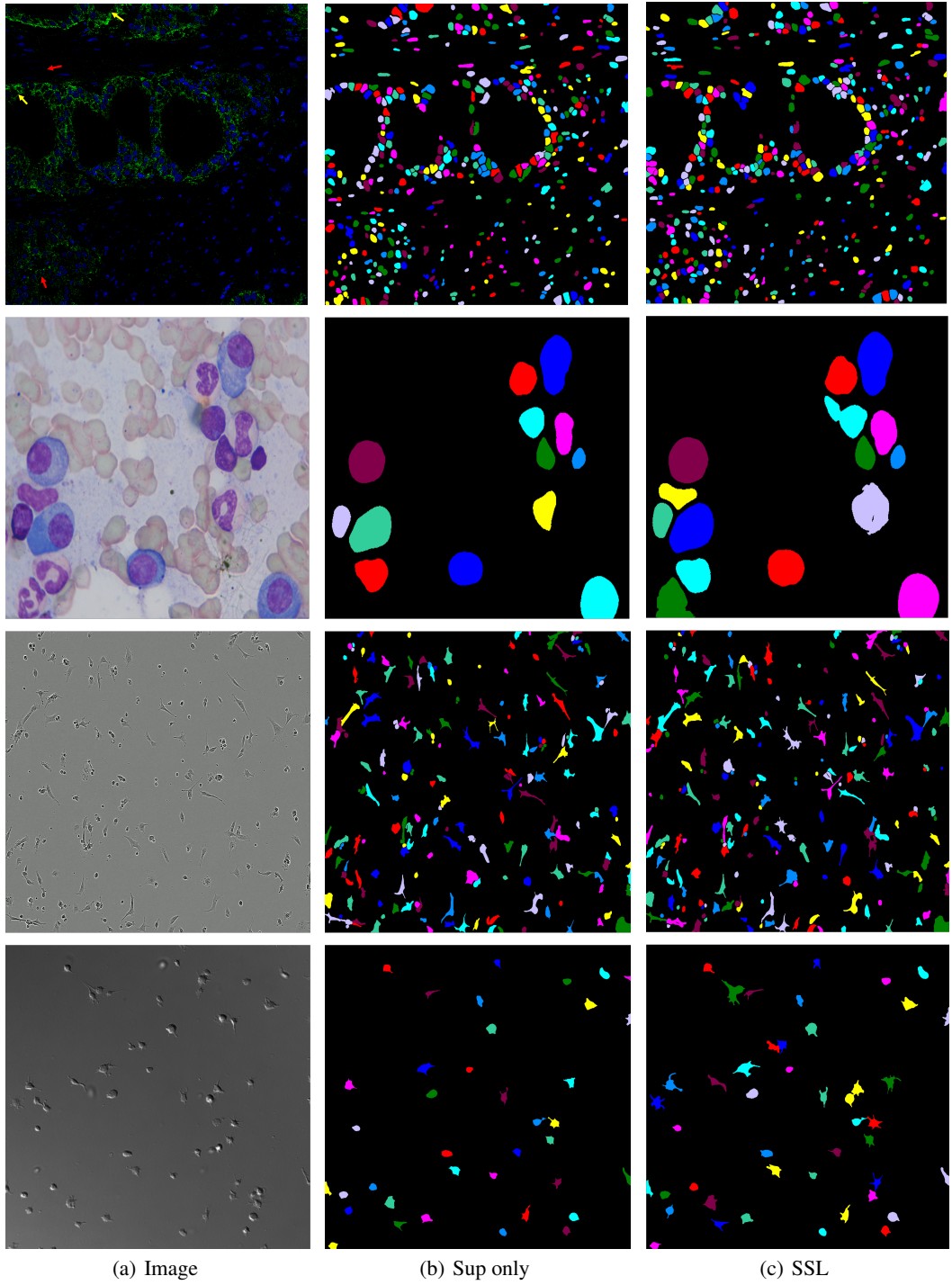

|                |                 |           |
| :------------: | :-------------: | :-------: |
| (a) Image      | (b) Sup only    | (c) SSL   |

Figure 4: Comparison of segmentation results between the supervised-only and semi-supervised models. "Sup only" denotes the supervised-only model trained with only labeled data, while "SSL" indicates the model trained with semi-supervised learning.

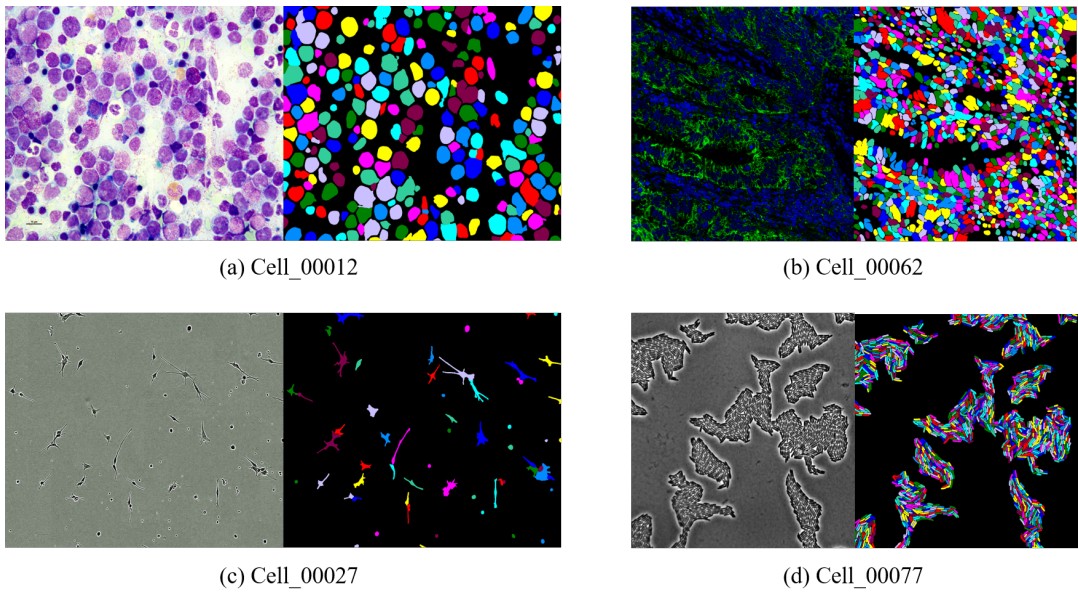

(a) Cell_00012            (b) Cell_00062

(c) Cell_00027            (d) Cell_00077

Figure 5: Visualized examples of successful (c and d) and failed cases (a and b).

## 4.3 Segmentation efficiency results on tuning set

Apart from the accuracy of the model, the inference speed of the model is also a key evaluation indicator. Therefore, we exploit several schemes to improve the inference speed while keeping the model accuracy. Specifically, we attempt to use different testing scales and window strides in the slide inference, as well as the half precision model and model pruning. The details of these experiments are shown in the Table 5. For multi-scale inference, we used (0.8, 1.0, 1.2) as the scale factors, except the fluorescence images whose scale factor is set as 1.5. For window stride in the slide inference process, we explore 4 different stride sizes and realize that as the stride increasing, the F1 score slightly decreases. Ultimately, the inference time of all cases in the tuning set is significantly smaller than the time tolerance, especially for the whole-slide image (case 101), our method only takes 758s, while the tolerance is 883s.

Table 5: The experiments of the running time. All the experiments test on the tuning set. (0.8,1.0,1.2)/1.5 indicates the 1.5 scale factor is adopted to the fluorescence images, while others keep using the (0.8,1.0,1.2) multi-scale fashion.

| Multi-Scale | Stride | Half | Pruning | Avg Running Time | F1 score |
|:---:|:---:|:---:|:---:|:---:|:---:|
| (1.0) | 256 | | | 11.98 | 0.8522 |
| (0.8,1.0,1.2) | 256 | | | 33.89 | 0.8504 |
| (0.8,1.0,1.2) | 360 | | | 19.58 | 0.8494 |
| (0.8,1.0,1.2) | 480 | | | 13.43 | 0.8493 |
| (0.8,1.0,1.2) | 512 | | | 12.39 | 0.8485 |
| (0.8,1.0,1.2) | 256 | ✓ | | 18.90 | 0.8505 |
| (0.8,1.0,1.2) | 256 | ✓ | ✓ | 12.75 | 0.8505 |
| (0.8,1.0,1.2)/1.5 | 256 | ✓ | ✓ | 12.49 | **0.8690** |

## 4.4 Results on final testing set

Results on the testing set are shown in Figure 6. Notably, our method ranks in the top 10 under the evaluation of F1-score and running time. However, there is a big variance in the performance on different modalities, shown in Table 6. Although our model shows good performance on the brightfield and phase-contrast cell images, it gives poor prediction of the differential interference contrast and the fluorescence cell images. Besides, the median F1 score is often bigger than the

mean F1 score for each modality. These indicate that our method is relatively poor in generalization, especially for fluorescent cell images. The reasons for this problem are supposed in two aspects. Firstly, there may be a large variance between the training and the testing set. Secondly, the scale factor of 1.5 especially for the fluorescence images is not suitable for the testing set, though it brings an improvement on tuning set.

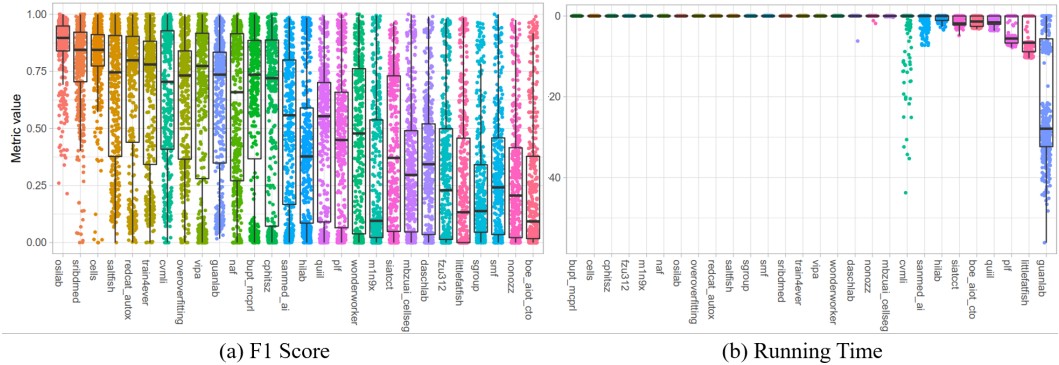

(a) F1 Score                                         (b) Running Time

Figure 6: The results on final testing set for each team. [18]

Table 6: The performance of MT$^2$ for each modality. DIC is the short for differential interference contrast.

| Modality | Brightfield | DIC | Fluorescent | Phase-contrast | All |
|---|---|---|---|---|---|
| Median F1 score | 0.8597 | 0.6217 | 0.0293 | 0.9019 | 0.7355 |
| Mean F1 score | 0.8534 | 0.5830 | 0.1704 | 0.8143 | 0.6085 |

### 4.5 Limitation and future work

The proposed method still has limitations. First of all, poor generalization on fluorescence images limits the performance of our model, which can be inferred from the results of the testing set. Second, the predictions of the deep learning model are not accurate enough. The prediction results of the model usually contain many small-sized false positive instances, which will greatly reduce the F1 score of instance segmentation. And the prediction scores for all pixels of a single cell are not smooth, so it is difficult to choose a suitable threshold. Besides, for brightfield and fluorescent images, the model suffers under- or over-segmentation problems, which can be alleviated by adopting sample balancing methods. Furthermore, the semi-supervised learning method we applied is merely the simplest mean teacher model, which will be optimized in the future.

## 5  Conclusion

In this paper, a novel semi-supervised cell segmentation method named Multi-task Mean Teacher (MT$^2$) is proposed to segment multi-modality microscopy images. It is based on the combination of Mean Teacher model and a multi-task learning method, in which two additional regression heads, i.e., the cell head and the boundary head, are introduced to further improve the prediction performance. Besides, a variety of strong data augmentations, a multi-scale inference, and an effective post-processing strategy also contribute to the promotion of the accuracy and generalization of our model. However, poor generalization on fluorescence images, false positives and under- or over-segmentation problems still exist, and the proposed method still has the potential to improve the efficiency in predicting the whole-slide images. Future work will be conducted to make above optimizations.

## Acknowledgement

The authors of this paper declare that the segmentation method they implemented for participation in the NeurIPS 2022 Cell Segmentation challenge has not used any private datasets other than those

provided by the organizers and the official external datasets and pretrained models. The proposed solution is fully automatic without any manual intervention.

This work is supported by National Key Research and Development Program of China (2022YFC0868500, 2020YFB2104604).

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
