# OpenReview forum: "MT2: Multi-task Mean Teacher for Semi-Supervised Cell Segmentation"
_NeurIPS.cc/2022/Challenge/CellSeg — Submitted to NeurIPS CellSeg 2022_

### Official Review · Reviewer_Zrz4 · 2022-12-19
**Paper Review**

**Rating:** 8
**Confidence:** 4

**Review:**

# Summary
This paper designed a semi-supervised framework named MT^2 for instance-level cell segmentation from images. The background and method details were demonstrated properly and the comparison and ablation experiments were presented clearly. I recommand the acceptance of this paper.

# Pros
1. This paper proposed a mean teacher method consisting teacher-student networks, which is suitable for the semi-supervised cell segmentation task with unlabeled data.
2. The loss function was designed properly with consideration of pixel classification, cell identification and boundary structures. The task of cell segmentation from images suffer from the overlapping of multiple cells and this paper explores a way to solve this issue.

# Cons
1. The post-processing step is important for segmentation task, especially for competitions. This paper also involves this step using the watershed method. Besides providing the thresholds of watershed method, the paper should introduce the principle or intuition of why to choose these thresholds. Also, it would be better to provide some visualization or quantitative evaluations.
2. As the paper stated in the limitation part, the false positive (FP) is an important issue in segmentation task. I suggest the authors provide the FP values alongside the F1-score for better illustration.

---

> ### Author Response · Authors · 2023-02-14
> **Response to Reviewer Zrz4**
>
> Thanks for your review.
> 1. We revised Section 2.3 to describe the procedures of our post-processing in more details, including the way we use watershed. Actually, we select the thresholds of the watershed algorithm based on the experimental results on Tuning Set as mentioned in Section 2.3. Besides, Figure 3 is introduced to visualize the effect of each step in post-processing for a better illustration.
> 2. Thanks for your advice. However, since the organizers do not give the ground-truth labels of the Tuning Set, we cannot calculate the FP values. So, we could only use the official F1-score for evaluation and discuss this issue in a qualitative way in Section 4.2.

---

### Official Review · Reviewer_Cz7H · 2022-12-25

**Rating:** 7
**Confidence:** 4

**Review:**

This paper proposed a multi-task Mean Teacher for semi-supervised cell segmentation. Extensive experiments show the superioriy of the proposed approach.

Pros:
1.  The overall architecture is simple yet effective. The notations and descriptions are easy to follow.
2.  The authors did additional research on extra cell data, which has proven the effectiveness in improving the segmentation performance.
3.  The proposed model achieved a decent trade-off between the accuracy and efficiency, as shown in Table 5.

Cons:
1.  I suggest the authors try different weak/strong augmentations beyond cutmix, as these augmentations are the key factors in semi-supervised settings. Also, it would be interesting to see some comparison results for different cell modalities as discussed by the authors
2. In Table 3, I think the order of these combinations should be changed as 'cutmix', 'color jitter', 'semi-supervised methods' are part of your main model, while elements such as 'post-process', 'multi-scale inference' do not belong to the model design. Better to re-organise them.

---

> ### Author Response · Authors · 2023-02-14
> **Response to Reviewer Cz7H**
>
> Thanks for your review.
> 1. In the experiments, we adopted the mainstream data augmentations in the current semi-supervised learning [1-6], i.e., random scale and crop, rotation, flipping for weak augmentations, and colorjitter, cutmix for strong ones. As these data augmentation methods have been proven effective and the ablation study of them is rarely included in these papers, we only applied directly without further exploration. Maybe we will try more data augmentations in the future work.
> Since the organizers do not offer the modality information of each image, and merely a whole F1-score on the Tuning Set can be obtained, it is difficult for us to provide the rigorous quantitative results of each modality. So, we could only observe and discuss the qualitative results in Figure 5 and Figure 4 as we added. Besides, we added the quantitative results for each modality on the test set in Table 6 and briefly discussed the variance model performance among the modalities.
> 2. Thank you for your suggestion. We have corrected Table 3 in a more reasonable way as you proposed.
>
> References: \
> [1] He R, Yang J, Qi X. Re-distributing biased pseudo labels for semi-supervised semantic segmentation: A baseline investigation[C]//Proceedings of the IEEE/CVF International Conference on Computer Vision. 2021: 6930-6940. \
> [2] Kwon D, Kwak S. Semi-supervised semantic segmentation with error localization network[C]//Proceedings of the IEEE/CVF Conference on Computer Vision and Pattern Recognition. 2022: 9957-9967. \
> [3] Fan J, Gao B, Jin H, et al. Ucc: Uncertainty guided cross-head co-training for semi-supervised semantic segmentation[C]//Proceedings of the IEEE/CVF Conference on Computer Vision and Pattern Recognition. 2022: 9947-9956. \
> [4] Guan D, Huang J, Xiao A, et al. Unbiased subclass regularization for semi-supervised semantic segmentation[C]//Proceedings of the IEEE/CVF Conference on Computer Vision and Pattern Recognition. 2022: 9968-9978. \
> [5] Yang L, Zhuo W, Qi L, et al. St++: Make self-training work better for semi-supervised semantic segmentation[C]//Proceedings of the IEEE/CVF Conference on Computer Vision and Pattern Recognition. 2022: 4268-4277. \
> [6] Liu S, Zhi S, Johns E, et al. Bootstrapping semantic segmentation with regional contrast[J]. arXiv preprint arXiv:2104.04465, 2021.

---

### Official Review · Reviewer_BgSE · 2022-12-27

**Rating:** 7
**Confidence:** 4

**Review:**

This paper proposed a multi-task Mean Teacher for semi-supervised cell segmentation. Experiments also evidenced the effectiveness of the proposed method.

Pros:
- The paper is well-written and easy to follow.
- The authors did a detailed study of how each part of the proposed method will affect the performance.
- Unlabeled images are relatively easy to obtain in practice, utilizing the unlabeled images to achieve better performance is great.

Cons:
- In the ablation study, it seems that SSL only contributes a little to the final score, thus, I hope the authors can give a more detailed study, for example, to show how SSL improves predictions.

---

> ### Author Response · Authors · 2023-02-14
> **Response to Reviewer BgSE**
>
> Thanks for your review.
> 1. The SSL results in Table 3 do indicate that the improvement of F1-score is limited. We suppose it may be due to a big difference between the unlabeled training set and the Tuning Set. In order to explain how semi-supervised learning improves the performance, Figure 4 in Section 4.2 is added to illustrate the comparison of segmentation results between the supervised-only and semi-supervised models. We have also summarized three merits of SSL in the revised paper. First, more positive samples can be predicted. Second, the predictions usually have finer cell boundaries. Third, fewer false positives are generated.

---

### Decision · Program_Chairs · 2023-01-19

Accept